# Vocal learning promotes patterned inhibitory connectivity

Mark N. Miller[1], Chung Yan J. Cheung[2] & Michael S. Brainard[1]

Skill learning is instantiated by changes to functional connectivity within premotor circuits, but whether the specificity of learning depends on structured changes to inhibitory circuitry remains unclear. We used slice electrophysiology to measure connectivity changes associated with song learning in the avian analog of primary motor cortex (robust nucleus of the arcopallium, RA) in Bengalese Finches. Before song learning, fast-spiking interneurons (FSIs) densely innervated glutamatergic projection neurons (PNs) with apparently random connectivity. After learning, there was a profound reduction in the overall strength and number of inhibitory connections, but this was accompanied by a more than two-fold enrichment in reciprocal FSI–PN connections. Moreover, in singing birds, we found that pharmacological manipulations of RA's inhibitory circuitry drove large shifts in learned vocal features, such as pitch and amplitude, without grossly disrupting the song. Our results indicate that skill learning establishes nonrandom inhibitory connectivity, and implicates this patterning in encoding specific features of learned movements.

[1] Howard Hughes Medical Institute and Departments of Physiology and Psychiatry, University of California-San Francisco, San Francisco, CA 94158, USA. [2] Neuroscience Graduate, Program, University of California-San Francisco, San Francisco, CA 94158, USA. Correspondence and requests for materials should be addressed to M.N.M. (email: millerm@phy.ucsf.edu)

Skilled motor behaviors including vocalizations are characterized by high degrees of precision, stereotypy, and adaptability, and are learned through practice that improves initially poor performance to a more expert level[1]. The precision and reliability of skilled behaviors is ultimately driven by highly structured neural activity in central premotor circuits[2–4], and connectivity patterns within premotor circuitry are critical for generating appropriate behavior. Nonrandom patterns of connectivity among excitatory neurons are a feature of many systems, and plasticity of specific excitatory connections is considered central to the capacity of networks to produce appropriate output[5–8]. However, whether learning shapes inhibitory connectivity to achieve comparable specificity[9,10] or instead promotes diffuse, nonspecific inhibition[11] is unclear. The development of temporally precise activation of a diffuse inhibitory network may be sufficient to structure premotor activity during vocal

**Fig. 1** Inhibition is a major component of RA circuitry. **a** Micrograph of acute RA slice preparation at three different magnifications showing RA with four patch electrodes during whole-cell recording (left), recorded and filled neurons visualized in the same slice (middle), and a group of two larger projection neurons (PNs) and two smaller inhibitory fast-spiking interneurons (FSIs, Int) from the middle panel's dotted region (right). PNs and FSIs are morphologically distinct. **b** PNs and FSIs express distinct firing patterns. PNs generate spontaneous tonic pacemaking activity at rest (left). After hyperpolarizing them with DC current, PNs produced adapting trains of high-amplitude (>30 mV) action potentials in response to depolarizing current steps, and they expressed voltage sag in response to further hyperpolarization (middle). In contrast, FSIs do not have spontaneous pacemaking activity, and they produce very high-frequency (>150 Hz) non-adapting trains of <30 mV action potentials in response to current injection (right). **c** Average action potential (AP) waveforms from an FSI (left) and a PN (right). **d** AHP amplitude and AP width across our sample of FSIs and PNs differentiated these cell types. **e** Inhibition dominates spontaneous synaptic activity in RA. Example spontaneous inhibitory postsynaptic currents (sIPSCs, top) and excitatory postsynaptic currents (sEPSCs, bottom) recorded from the same PN at equal distance from either $E_{Cl}$ or $E_{AMPAR}$. Both the frequency and amplitude of sIPSCs are much larger than those of sEPSCs despite the pacemaking activity of other PNs in the circuit. **f** Average excitatory vs inhibitory charge recorded from 46 PNs demonstrates that inhibition outweighs excitation in RA PNs during spontaneous activity

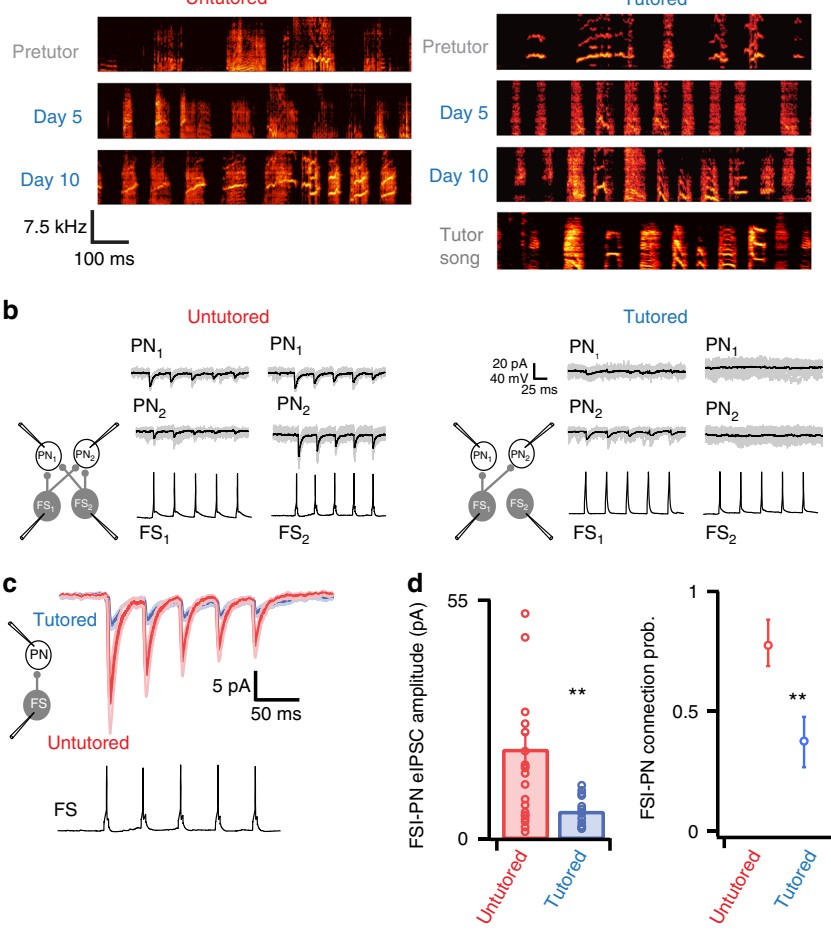

**Fig. 2** Learning to copy a tutor song is associated with dramatic changes to RA inhibitory strength and connectivity. **a** Tutoring Bengalese Finches with an automated tutoring paradigm drives song learning in 10–20 days. Spectrograms from age-matched untutored birds show that they retain unstructured subsong-like vocalizations over this period (left), whereas spectrograms from tutored birds (right) rapidly converge on a copy of the tutor stimulus (bottom panel). The same stimulus was used to tutor all birds so that they ultimately produced similar vocal output at the time of slice recording. **b** Examples of evoked-IPSCs from paired recordings of PNs and FSIs from untutored (left) and tutored (right) birds. Driving single spikes in FSIs (bottom traces) produced unitary IPSCs in synaptically connected PNs. Gray traces are individual current sweeps and black traces are averages of 15–30 sweeps. IPSCs were large, reliable, and depressing ($0.63 \pm 0.08$ at 50 Hz) in connected FSI→PN pairs. These features are consistent with the high level of spontaneous inhibition that we observed in sIPSC recordings. **c** After song learning, FSI→PN connections were significantly weaker than that in untutored birds. Blue trace is mean IPSC ± SEM from tutored birds ($n = 24$ pairs from 11 birds) and red trace is mean IPSC ± S.E.M. from untutored birds ($n = 47$ pairs from nine birds). Asterisks indicate significance of $p < 0.01$ determined by $t$-test. **d** IPSC amplitude was significantly smaller in tutored birds than in untutored birds (left), and the FSI→PN connection probability was reduced from 0.76 to 0.38 after tutoring. Error bars on the right plot indicate 95% confidence intervals of connection probabilities from a binomial distribution. Asterisks indicate significance of $p < 0.01$ determined by binomial test

learning[12], yet formation of specific inhibitory connectivity in simulated networks is also sufficient to stably encode complex activity patterns[13]. This motivated us to ask how learning shapes inhibitory connectivity in songbirds, where robust vocal learning is subserved by a well-characterized premotor network.

We used a slice preparation of the avian vocal premotor nucleus RA (Fig. 1a) to examine changes to motor circuitry over the course of vocal learning. Glutamatergic RA projection neurons (PNs) that innervate vocal and respiratory motoneurons[14] produce highly structured activity[2,3] that emerges during learning[15] and ultimately controls the acoustic features of the learned song[4]. Previous studies that manipulated local RA circuitry reported minimal effects on production of learned song, and raised the possibility that moment-by-moment activity in RA projection neurons is largely determined by excitatory input from RA's afferent regions, HVC and LMAN[16,17]. However, within RA, GABAergic FSIs innervate PNs and can coordinate ongoing activity across PNs in acute slices[18]. Moreover, in other systems,

FSIs can potently modulate ongoing neural activity patterns[19–23]. These findings raise the possibility that FSIs might be critical modulators of RA activity that undergo plasticity during song acquisition and contribute to the encoding of learned song features. We therefore carried out experiments to test whether connectivity between FSIs and PNs within RA is shaped during vocal learning, and how inhibitory circuitry within RA contributes to the control of learned vocalizations.

## Results

We first established that RA cell types in BFs can be differentiated by their spontaneous and evoked firing properties in slices[18] (Fig. 1). PNs are spontaneously pacemaking, whereas FSIs are only sporadically active, and PNs produce adapting trains of action potentials in response to current injection, whereas FSIs produce high-frequency (>200 Hz) trains of short, narrow spikes with large afterhyperpolarizations and minimal spike-frequency

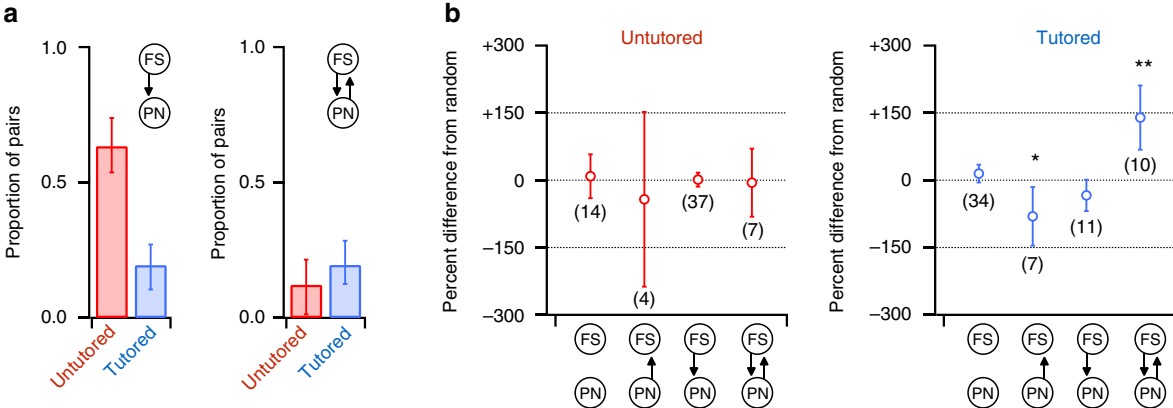

**Fig. 3** Enrichment of reciprocal FSI ↔ PN connections during song learning. **a** Proportion of all tested pairs before (red) and after (blue) tutoring that were unidirectional FSI→PN connections (left) or reciprocal FSI ↔ PN connections (right). Song learning was associated with a 69% reduction in unidirectional FSI→PN connections (binomial test $p < 0.001$), but did not alter the proportion of reciprocal FSI ↔ PN connections. Error bars indicate 95% confidence intervals. **b** Frequency of first-order connection patterns that we detected in untutored (left, red) and tutored (right, blue) birds relative to each connection pattern's expected frequency given unidirectional FSI→PN and PN→FSI connection probabilities in untutored and tutored birds. Each connection pattern was present at chance levels before tutoring, whereas reciprocal FSI ↔ PN connections were 2.39 times more frequent than predicted by random connectivity after song learning (multinomial test $p < 0.01$, 95% confidence interval = 1.68–3.11-fold, see Methods). Correspondingly, one-way PN→FSI connections were 0.41-fold less frequent than predicted after song learning (multinomial test $p < 0.05$). Relative to untutored birds, tutoring significantly increased the prevalence of FSI ↔ PN and PN→FSI connections (multinomial test $p < 0.005$) and reduced the prevalence of FSI→PN connections (multinomial test $p < 0.00001$). Error bars indicate 95% confidence intervals. Connection pattern sample sizes are parenthetical

adaptation (Fig. 1b–d, Supplementary Fig. 1). PNs and FSIs are thought to form both homotypic and heterotypic synapses and also receive excitatory inputs from HVC and LMAN[24–26]. Because the prevalence, pattern, and strength of these different connections determines how RA transforms its input into activity patterns that drive structured vocal output during song, we next sought a general description of synaptic activity patterns within RA that might contribute to RA function.

Due to PN pacemaking (Fig. 1b), RA slices are highly spontaneously active even without imposing conditions that enhance excitability[24] (Fig. 1e). This activity state mirrors RA activity in vivo under anesthesia and during awake non-singing states[2–4]. We exploited the correspondence between in vivo and in vitro activity states to monitor ongoing synaptic inputs to PNs in acute RA slices from adult (p90-180) male BFs. Spontaneous excitatory and inhibitory synaptic currents (sEPSCs and sIPSCs) on PNs were recorded in voltage clamp with cesium and QX-314 in the pipette solution to permit isolation of excitatory and inhibitory synaptic currents via manipulation of the holding potential. PNs received both spontaneous excitatory and inhibitory synaptic currents (Fig. 1e) under these conditions. Despite the strong pacemaking activity of RA PNs, we found that PNs receive much more spontaneous inhibition than excitation (Fig. 1e, f), consistent with previous reports[18]. Inhibitory input to PNs was 6.18-fold greater than excitatory input (SEM = 0.53-fold, $p < 1^{-12}$), and we found no examples of PNs receiving more excitation than inhibition (Fig. 1f).

Since inhibition dominates synaptic activity within RA (Fig. 1f) and is important for the production of structured activity in other systems, we asked whether the inhibitory circuit in RA is a locus of plasticity during vocal learning by comparing RA inhibitory synapses between a group of untutored birds and a group of birds that had completed song learning. We used a computerized tutoring paradigm[27] to teach p40–45 BFs an identical tutor song, and maintained other age-matched birds as an untutored comparison group. The ages of tutored and untutored birds were not significantly different (untutored age = 66.1 ± 8.6 SD days, tutored age = 69.7 ± 9.2 SD days, $p = 0.37$, Supplementary Fig. 2). Tutored birds learned to copy the tutor song within 2–3 weeks,

while untutored birds continued to produce unstructured juvenile vocalizations (Fig. 2a, Supplementary Fig. 3). To characterize changes to inhibitory circuitry during learning, we measured synaptic connections between PNs and FSIs using simultaneous whole-cell recordings in acute RA slices (Fig. 2b). Tutoring had a profound effect on RA inhibitory synapses: unitary FSI→PN IPSCs were on average 3.23-fold weaker in tutored than in untutored birds of the same age range (IPSC$_{untut}$ = 21 ± 3.5 pA; IPSC$_{tut}$ = 6.5 ± 0.5 pA; $p < 0.01$, Fig. 2c, d). Furthermore, the probability of FSI→PN connections ($P_C$) was reduced by nearly 50% after tutoring (untutored $P_C$ = 0.76, 47 connections out of 62 tested; tutored $P_C$ = 0.4, 24 connections out of 60 tested; binomial test $p < 0.01$, Fig. 2d). Overall, these data indicate that FSI→PN connections are a major feature of RA circuitry that undergo dramatic pruning and weakening during song learning.

Reduction of FSI→PN $P_C$ during song learning could reflect indiscriminate loss of random FSI connections, or it could reflect selective rewiring that preserves or creates functionally important subcircuits within RA. To investigate these possibilities, we separately examined changes to the proportion of pairs in which there was a unidirectional connection from an FSI to PN (FSI→PN) and the proportion of pairs in which there were reciprocal connections between an FSI and PN (FSI ↔ PN). Random loss of FSI connections would produce a decrease in the proportion of both of these patterns. We indeed found that the proportion of unidirectional FSI→PN pairs decreased by 69% between untutored and tutored birds (binomial test $p < 0.001$ Fig. 3a, left). In contrast, the proportion of reciprocally connected FSI ↔ PN pairs remained constant over tutoring (Fig. 3a, right). This increase in the relative proportion of reciprocal FSI ↔ PN pairs despite an overall pruning of inhibitory connections suggests a nonrandom process that preferentially preserves or creates reciprocal connectivity between FSIs and PNs while eliminating the majority of unidirectional FSI→PN connections.

We further tested this possibility by investigating whether the frequency of different patterns of connectivity between FSIs and PNs exhibited any deviations from random in our tutored and untutored paired-recording data sets (Fig. 3b). We considered the

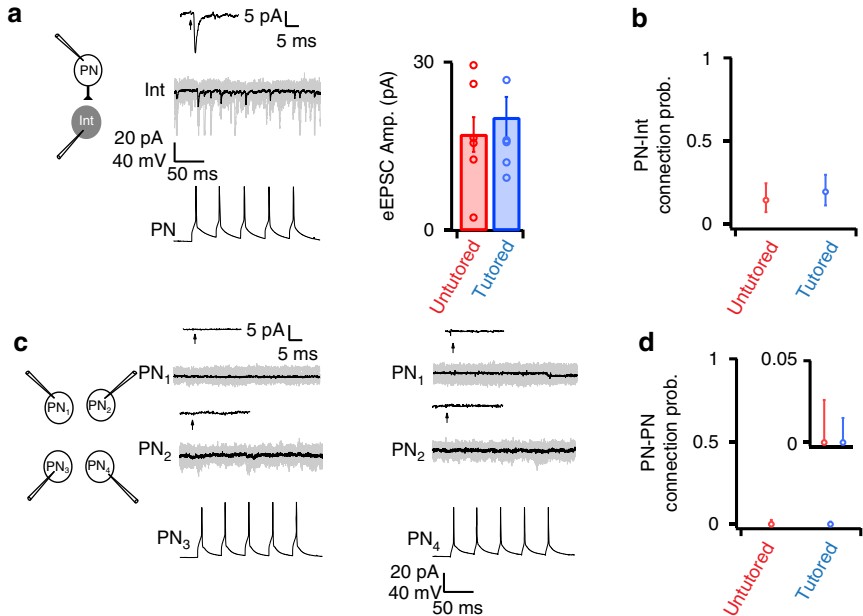

**Fig. 4** Excitatory PN synapses were not altered during song learning. **a** Example PN→FSI connection. Evoked PN spikes (bottom) drove monosynaptic EPSCs in a postsynaptic FSI (middle). The black trace is the average of 30 individual sweeps (overlaid gray traces). PN→FSI connections exhibited modest short-term depression (0.63 ± 0.06 at 50 Hz), as observed in other systems[41]. The black trace above the sweeps is the spike-triggered average (STA) of 375 events to enhance sensitivity to small EPSCs. The arrow indicates the presynaptic spike time. PN→FSI strength did not change during song learning (right panel, $n = 10$ pairs from 3 untutored birds, $n = 17$ pairs from five tutored birds). **b** Tutoring did not alter the probability of PN→FSI connections (0.19 untutored, 0.17 tutored). Error bars indicate 95% confidence intervals from a binomial distribution. **c** Example of quadruple recording from four PNs. Brief current pulses evoked spikes in one PN (bottom), while the other PNs were voltage-clamped at $E_{Cl}$ to detect evoked EPSCs. Gray current traces are 75 overlaid single sweeps and the black trace is the average. We were unable to detect any PN–PN connections in 262 attempts. **d** Summary of attempts to record PN–PN connections in untutored (red) and tutored (blue) birds. The upper 95% confidence interval is 0.026 in untutored birds and 0.014 in tutored birds

four possible patterns of connections between a given FSI and PN: (1) no connection, (2) unidirectional connection from FSI to PN, (3) unidirectional connection from PN to FSI, or (4) reciprocal connection between FSI and PN. To test whether these connectivity patterns deviated from random, we created separate null models (see Methods) for tutored and untutored data sets that established how prevalent each connection pattern between FSIs and PNs would be if there were no specific patterning beyond that arising from the measured probabilities of unidirectional connections ($P_C$ for FSI→PN and $P_C$ for PN→FSI). In untutored birds, all connection patterns were observed at chance levels, consistent with an initially random patterning of connections between FSIs and PNs (Fig. 3b, left). However, in tutored birds, reciprocal FSI ↔ PN patterns were present at more than double the probability expected by chance (multinomial test $p < 0.01$, Fig. 3b, right), and the proportion of FSI ↔ PN connections among all connections in tutored birds was significantly greater than in untutored birds (multinomial test $p < 0.005$).

This indicates that learning promotes specific, nonrandom rewiring of RA circuitry by sparing or creating reciprocal FSI ↔ PN connections, even as overall inhibitory connectivity is reduced, resulting in a network that is enriched in reciprocal connections between FSIs and PNs.

In contrast to the strong effects of tutoring on FSI→PN connection probability, strength, and patterning (Figs. 2 and 3), tutoring had no detectable effect on PN excitatory connections within RA (Fig. 4). We encountered excitatory PN→FSI connections less frequently than FSI→PN connections, and tutoring did not alter PN→FSI EPSC amplitude (Fig. 4a, $EPSC_{untut} = 17.1 \pm 3.1$ pA, $EPSC_{tut} = 20.1 \pm 3.7$ pA, $p = 0.63$) or connection probability (Fig. 4b, untutored $P_C = 0.17$, tutored $P_C = 0.19$, binomial

test $p = 0.73$). Unlike PNs, FSIs receive high levels of spontaneous excitatory input (Fig. 4a, overlaid gray traces), presumably from presynaptic PN pacemaking activity. To confirm PN→FSI connections in this background activity, we used the spike-triggered average EPSC evoked by 100–250 PN spikes (Fig. 4a, Supplementary Fig. 4) to evaluate synaptic connections and calculate PN→FSI $P_C$. In addition to PN→FSI connections, we also used spike-triggered average EPSCs to search for PN–PN synapses, because they are suggested to play important roles in song patterning and learning[25]. However, we were unable to detect any PN–PN connections in 262 attempts, indicating that under our conditions these synapses are either very rare ($P_C < 0.02$, Fig. 4d), very weak ($g_{syn} < 6.25$ pS, Fig. 4c), or both.

Our observations that FSI connectivity is a primary substrate for intrinsic interactions within RA (Figs. 1, 2, and 4) and that FSI→PN synapses are a major locus of plasticity during song learning (Figs. 2 and 4) led us to examine the functional contribution of FSIs to song production. Because we found that FSI connectivity gains specificity in parallel with the acquisition of highly structured learned vocal output (Fig. 3), we were specifically interested in the possibility that FSI activity is critical for producing learned acoustic features during singing. Patterned bursts across the RA PN population are thought to drive acoustic features including the fundamental frequency (FF) and amplitude of each vocalization[2–4], and prevailing models of song production hold that RA PN burst patterns are inherited from afferent HVC inputs to PNs[3,16,28]. However, potential roles for RA inhibitory circuitry in shaping song production have not been examined, even as inhibitory circuits are known to critically shape patterned activity in other systems[29], including songbird HVC[12]. We tested whether RA FSIs contribute to the control of learned song

features by pharmacologically manipulating RA inhibitory circuitry in singing birds and measuring the effects on the acoustic structure of learned song.

To decrease RA inhibitory function, we used 1-Naphthyl acetyl spermine (NASPM) to block glutamatergic excitatory inputs to FSIs. In many systems, glutamatergic inputs to FSIs are primarily mediated by AMPARs that lack the gluA2 subunit[30,31], and NASPM specifically antagonizes these gluA2-lacking receptors[32]. NASPM can therefore reduce recruitment of FSIs and decrease inhibitory gain without completely blocking GABAergic transmission, which might produce pathological activity states. We confirmed in RA slices from adult BFs that bath application of 0.1 mM NASPM attenuated the overall level of spontaneous inhibition received by PNs by 42% ($p = 0.0009$, Fig. 5a, b),

indicating that NASPM is an effective tool to reduce RA inhibitory function.

We next asked if RA inhibitory circuitry contributes to the production of learned song features by delivering NASPM (1–2 mM) into RA with reverse-microdialysis in vivo during singing[33,34]. We measured NASPM's effect on the FF and amplitude of song syllables, because these are features that are learned from the tutor song that are subsequently maintained within a narrow range for the lifetime of the bird[27,35,36]. NASPM robustly increased both FF ($4.4 \pm 0.7\%$ SEM, $n = 12$, $p = 0.0014$, Fig. 5d, e) and syllable amplitude ($74.4 \pm 18.9\%$ SEM, $n = 12$, $p = 0.0008$, Fig. 5f, g) without altering overall syllable structure or otherwise disrupting the song (Fig. 5c), indicating that RA inhibitory circuitry can potently regulate the magnitude of specific learned syllable features during singing.

If inhibitory gain directly regulates learned vocal features and suppressing inhibitory function with NASPM increases syllable FF and amplitude, enhancing RA inhibitory function should produce the opposite effects. We tested this prediction by pharmacologically enhancing RA GABAergic function by reverse-microdialysis of the benzodiazepine midazolam, which allosterically increases the open probability of ligand-bound $GABA_AR$. In contrast to NASPM, midazolam (2.5 mM) reliably and significantly reduced both FF ($-2.9 \pm 0.7\%$, $n = 5$, $p = 0.021$, Fig. 5d, e) and syllable amplitude ($-34.4 \pm 4.5\%$, $n = 5$, $p = 0.004$ Fig. 5f, g). Like NASPM, however, midazolam dialysis specifically altered FF and amplitude without altering syllable structure (Fig. 5c), indicating that neither drug grossly disrupted the overall pattern of RA activity, but instead modulated RA activity in a specific fashion that shifted FF and amplitude.

## Discussion

Our results show that RA inhibitory circuitry is a major locus of plasticity during song learning and that inhibition is a dominant component of RA circuitry that controls the production of

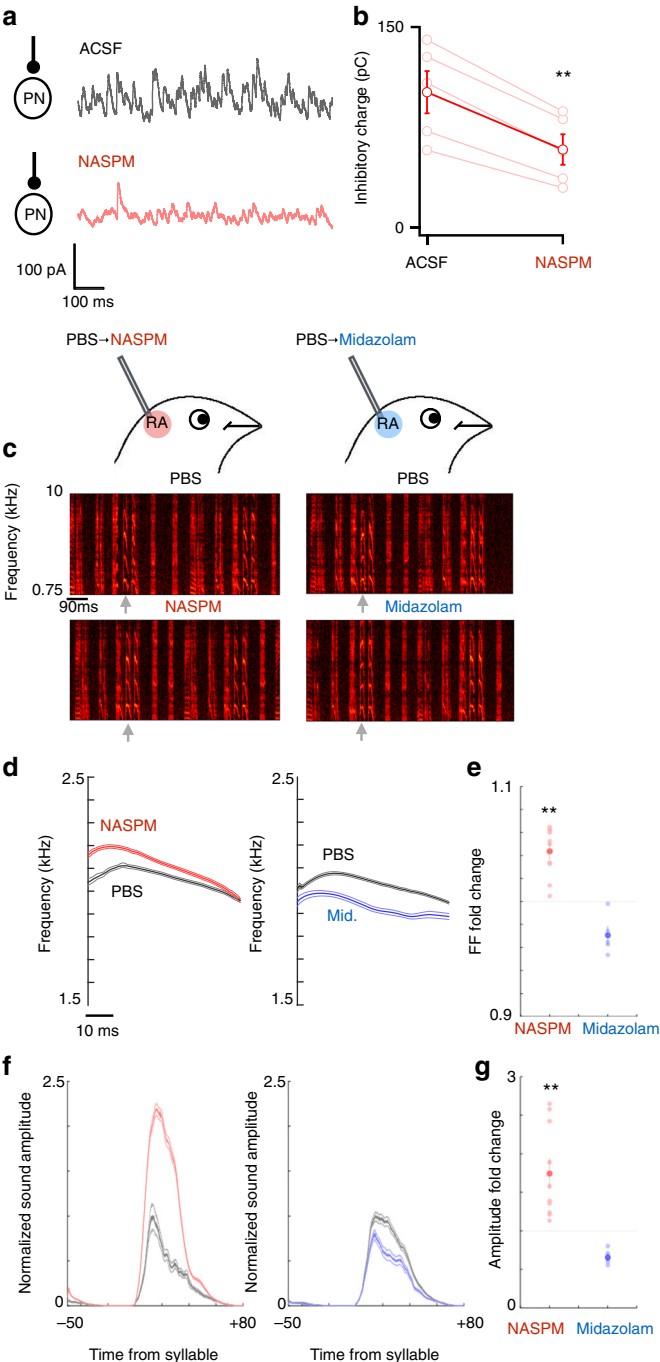

**Fig. 5** Manipulating RA inhibition in singing birds bidirectionally shifts learned song features, while preserving overall song structure. **a** NASPM reduces spontaneous inhibition in RA slices. Example sIPSCs recorded from a PN in ACSF (gray trace) and after 0.1 mM bath application of NASPM (red trace). **b** NASPM reduced spontaneous inhibition in all PNs recorded. On average, NASPM reduced inhibitory charge by 42%. **c** Schematic of in vivo reverse-microdialysis experiment and example spectrograms of songs recorded during reverse-microdialysis of PBS followed by either NASPM (left), which reduces RA inhibition, or midazolam (right), which enhances RA inhibition. Overall song structure was not altered by either manipulation. The gray arrow indicates the syllable that we analyzed for fundamental frequency (FF) and amplitude in **d**–**g**. **d** Fundamental frequency contours of the syllable highlighted by the gray arrow in **c** during PBS dialysis (gray) followed by either NASPM (red) or midazolam (blue) dialysis. Envelopes indicate SEM. Reducing RA inhibition with NASPM increased the FF without altering syllable structure (left), whereas enhancing RA inhibition with midazolam reduced the FF of the syllable (right). **e** Changes in syllable FF induced by either NASPM (red) or midazolam (blue) across all experiments. Light circles are fold-changes in FF from PBS in individual experiments and darker circles indicate each condition's mean. Error bars indicate SEM. **f** Rectified amplitude waveforms of the syllable marked by gray arrows in **c** during PBS (gray) followed by either NASPM (red) or midazolam (blue) dialysis. Waveforms are normalized to the mean of syllables produced during PBS dialysis. Envelopes indicate SEM. NASPM significantly increased syllable amplitude, whereas midazolam decreased it. **g** Change in syllable amplitude relative to PBS induced by either NASPM (red) or midazolam (blue) across all experiments. Light circles indicate individual experiments and darker circles indicate condition means. Error bars indicate SEM

learned vocal features. RA is a cortical analog that projects to brainstem premotor nuclei innervating vocal and respiratory musculature[37], and patterned activity in RA projection neurons is widely presumed to participate in the moment-by-moment control of learned features of song[2–4,38]. Previous work has focused on the excitatory inputs to RA from HVC and LMAN as primary sites of synaptic plasticity responsible for establishing patterned activity within RA during song learning, and thereby encoding learned features of song[26,28,39]. Despite the importance of inhibitory function within the upstream vocal motor region HVC for song learning and production[12,23] and indications that interneurons in RA are capable of potently controlling circuit activity in vitro[18], potential roles for RA inhibitory circuits in song production and learning have received little attention. Here, we show that song learning is associated with dramatic pruning of local inhibitory circuitry within RA (Fig. 2), and that this pruning remodels initially random inhibitory connections to selectively preserve reciprocal projections between fast-spiking interneurons and projection neurons (Fig. 3). Moreover, we demonstrate that manipulating inhibitory function within RA of singing birds can drive bidirectional changes to learned acoustic features of song (Fig. 5). Together, these data indicate that encoding of learned song features depends on inhibitory function in RA that is sculpted during song learning, and is not simply inherited from patterns of HVC afferent activity.

Our finding that RA FSI→PN connectivity is initially widespread and nonspecific and then becomes enriched in specific reciprocal patterns during vocal learning (Fig. 3) indicates that acquisition of learned skills may rely in part on the formation of specific patterns of inhibitory connections in addition to plasticity of excitatory connections. This result raises the possibility that diffuse inhibitory connectivity found in neocortex[11] may reflect a substrate for learning that has not yet occurred, and that specific inhibitory connectivity patterns in other systems[9,10] may similarly be a product of inhibitory circuit plasticity during learning. Consistent with the possibility that inhibitory circuitry is shaped during learning to encode specific song features, we found that modulating inhibitory gain in RA of singing birds alters the precisely controlled values of FF and amplitude that are learned during song acquisition. This suggests that vocal features are encoded in premotor inhibitory networks during learning, and inhibitory activity in RA subsequently controls the production of these features.

The sculpting of inhibitory circuitry that we describe here likely interacts with other circuit modifications to encode the learned song. HVC inputs to RA PNs are also pruned during song learning[28] suggesting that vocal learning engages multiple processes to reduce shared synaptic inputs to RA PNs. Shared inputs including initially exuberant and powerful FSI→PN connectivity might prevent different groups of PNs from independently varying, thereby limiting the complexity and precision of vocal output. Hence, one function of diminished FSI→PN connections during learning might be to enable the formation of sparser and more independently varying PN ensembles required to control the acoustic features of learned song. Additionally, the experience-driven enrichment of reciprocal FSI ↔ PN connections that we observed might be particularly important for generating RA's characteristically precise premotor activity patterns[2–4], which gradually emerge over song learning[15] and are thought to be critical for the moment-by-moment control of syllable FF and amplitude.

More generally, longstanding models attribute control of acoustic features such as FF and syllable amplitude to RA activity on the indirect basis of anatomy[37], RA activity patterns[2–4], and the disruptive effects of electrically stimulating RA[38]. Here we provide a causal demonstration that bidirectional manipulation of

inhibition in RA produces corresponding bidirectional changes in FF and amplitude. These results further establish RA as a primary source of control signals for learned acoustic features of song, and additionally provide insight into the nature of those control signals: they support a model in which increased activity across the population of PNs (associated with a decrease in inhibitory tone) drives an increase in vocal and respiratory muscle tensions, and corresponding increase in FF and amplitude, while a decrease in PN firing (associated with an increase in inhibitory tone) results in a decrease in FF and amplitude. Together with our finding that song learning is associated with profound remodeling and increased specificity of RA inhibitory connections, these results from singing birds suggest that the specific pattern and strength of inhibitory connections within RA that are shaped during song acquisition determines the precise values of FF and amplitude produced during learned song.

## Methods

**Animals.** Data from 39 male Bengalese Finches are included in this study. All birds were from our breeding colony at UCSF, and experiments were conducted in accordance with NIH and UCSF policies governing animal use and welfare.

**Song tutoring.** We adapted a computerized tutoring protocol[27] to provide finches with a common learning environment, equal exposure to tutor stimuli, and to explicitly constrain the period over which learning could occur. Clutches of Bengalese Finches from our breeding colony were raised from eggs by foster females (2–3 per nest) in sound proof chambers (Acoustic Systems) to prevent exposure to male songs or other tutor stimuli throughout early development. At 35 days post hatch (p35), we transferred each male bird to individual housing within individual sound proof chambers on a 14/10 h light/dark cycle. At p40–p45, we initiated tutoring by activating an operant perch in each cage that triggered tutor song playback through a speaker in the chamber (75 dB). Untutored birds were housed in identical cages, except that their perch triggers were inactive. All tutored birds were tutored with an identical stimulus that we constructed from seven acoustically distinct syllables and two intro notes chosen from a library of recorded Bengalese Finch vocalizations, separated by inter-syllable gaps drawn from the distribution of gaps produced by finches in our colony. Each perch-triggered playback consisted of three identical renditions of the tutor stimulus. We limited playbacks to 3 sets of 10 per day because we found that ad-lib playbacks prevented good learning, as previously reported[27]. Vocalizations were detected and recorded with custom LabView software. Once birds learned to produce a copy of the tutor stimulus, they were taken from the tutoring apparatus for slice preparation. We usually prepared slices from tutored and untutored birds on consecutive days to achieve age-matching across conditions. Tutored birds that failed to copy the tutor song, retained unstructured juvenile vocalizations, or produced stereotyped song that was different from the tutor song were not included in electrophysiology experiments.

**Electrophysiology.** RA slices were prepared as previously described[26]. The birds were deeply anesthetized with 4% isoflurane and decapitated in ice-cold oxygenated ACSF containing 125 mM Choline-Cl, 2.5 mM KCl, 2 mM MgCl$_2$, 1.25 mM NaHPO$_4$, 26 mM NaHCO$_3$, and 1 mM CaCl$_2$, and adjusted to 350 mOsm with dextrose. 250um coronal or sagittal RA slices were cut (Leica VT1000S) from each hemisphere under cold, oxygenated ACSF and transferred to an interface holding chamber with 38 °C recording ACSF containing 125 mM Choline-Cl, 2.5 mM KCl, 2 mM MgCl$_2$, 1.25 mM NaHPO$_4$, 25 mM NaHCO$_3$, and 2 mM CaCl$_2$, and adjusted to 350 mOsm with dextrose. After 30 min, slices in the holding chamber were relaxed to room temperature. During recording, bath temperature was maintained at 38 °C with a feedback-controlled inline heater (Warner Instruments). The slices containing RA were submerged in ACSF on the stage of an Olympus BX-51WI microscope and RA was identified with a ×4 or ×10 objective. Neurons in RA were visualized with DIC optics using a ×40 water-immersion objective. Patch pipettes were pulled on a Sutter P-97 puller to achieve tip impedances of 4–10 MΩ. To record spontaneous E/IPSCs, pipette solution contained 20 mM KCl, 100 mM Cs-MethylSulphonate, 10 mM K-HEPES, 0.1% biocytin, 4 mM Mg-ATP, 0.3 mM Na-GTP, 10 mM Na-Phosphocreatine, and 3 mM QX-314, with pH 7.35, and was adjusted to 315 mOsm with sucrose. For paired recordings that required intact action potential generation, pipette solution contained 20 mM KCl, 100mM K-gluconate, 10mM K-HEPES, 0.1% biocytin, 4 mM Mg-ATP, 0.3 mM Na-GTP, and 10 mM Na-phosphocreatine, with pH 7.35, and was adjusted to 315 mOsm with sucrose. Whole-cell recordings from PNs and FSIs were obtained under visual guidance with a ×40 water-immersion objective, current or voltage records were amplified by Multiclamp 700B (Molecular Devices) or Axopatch 1 C/1D (Axon Instruments) amplifiers, digitized at 10 kHz, and recorded with custom IGOR Pro software (Wavemetrics). Pipette capacitance and series resistance were compensated online and series resistance was monitored at 2 min intervals. Recordings with series resistance >20 MΩ or monotonic 25% change in input resistance were

discarded. PNs were distinguished from FS interneurons in loose-patch mode on the basis of PN spontaneous pacemaking activity, by post hoc inspection of bio-cytin fills, by FSIs lower input resistance, and by differences in action potential shape AHP and width (Fig. 1c, d) when using K-Gluconate pipette solution. Action potential amplitides reported in Fig. 1 are relative to 0 mV. Spontaneous IPSCs were recorded in voltage clamp as outward currents at the measured mixed-cation reversal potential determined by reversing sEPSCs, and spontaneous EPSCs were recorded as inward currents at $E_{Cl}$ determined by observing the reversal of sIPSCs. Reversal potentials were always within 5 mV of the calculated reversal potential when corrected for the liquid junction potential measured during each experiment. Paired recordings were made from 2–4 simultaneously recorded neurons. We tested for synaptic connections between neurons by driving 100–250 action potentials in each neuron with 1–2 ms 0.3–1 nA pulses at 50 Hz with 10 s duty cycle, while monitoring synaptic responses in other simultaneously recorded neurons in voltage clamp at −70 mV. Both inhibitory and excitatory connections were apparent in averaged sweeps, but we also calculated the spike-triggered average offline for all potential synaptic partners to increase sensitivity to very weak connections (Supplementary Fig. 4). We never detected a connection with the spike-triggered average that was not also detected in the averaged sweeps. To maximize our sample of FSIs, which are a minority of neurons in RA, we intentionally targeted small neurons for recording until we found an FSI. Our sample of FSIs, therefore, had smaller somata than our sample of PNs on average. However, we also encountered FSIs with somata as large as PNs, consistent with Spiro et al.[18], which found overlapping distributions of PN and FSI soma size.

**Connection motif analysis**. We built separate models for untutored and tutored data sets by creating networks with random connectivity based on the unidirectional (FSI→PN, PN→FSI, and PN → PN) connection probabilities that we measured with paired recordings in each condition. To calculate the likelihood of observing reciprocal FSI–PN connections at the rate present in our data sets, we simulated tutored and untutored networks constructed with each data set's unidirectional connection probabilities and sample size 100,000 times. Conceptually, this approach extends a binomial test to an arbitrary number of potential outcomes (in our case, the four possible connection motifs). To test whether motifs in tutored birds were significantly more or less common than in untutored birds, we created models with the frequencies of each connection pattern (FSI→PN, PN→FSI, FSI ↔ PN) present in untutored birds, and calculated the likelihood of observing the frequencies present in tutored birds. We validated our models with Matlab's mnpdf () and mnrnd() functions.

**In vivo reverse-microdialysis**. We pharmacologically manipulated RA in vivo in freely behaving and singing birds as previously described[33,34]. Adult (>100 days post hatch) male bengalese finches were implanted bilaterally with microdialysis probes (CMA) targeted to RA. Accurate placement in RA was confirmed during surgery by extracellular recording of RA's characteristic spontaneous activity. After recovery from surgery, the birds were housed individually within sound attenuating chambers (Acoustic Systems) on a 14/10 h light/dark schedule with free access to food, grit, and water, and vocalizations were recorded with a microphone fixed to the cage ceiling. PBS was continuously delivered to RA at a rate of 0.1 μl/min via a fluid commutator connected to a syringe pump outside the bird's isolation chamber. To manipulate inhibitory function within RA, we switched from PBS to either NASPM (Tocris) or midazolam (Sigma). Because the switch occurred outside the isolation chamber, the birds remained undisturbed and continued to behave and sing normally through the transition from PBS to drug. A total of >100 undirected song bouts were collected during both PBS and subsequent drug dialysis for each experiment.

**Acoustic feature analysis**. The songs were recorded at 32 kHz and 16-bit depth with custom LabView software[33,34,36]. Offline, the syllables were extracted from audio files based on amplitude threshold crossings of the rectified audio waveform smoothed with a 2 ms moving window and analyzed using custom Matlab software. We focused on syllables with prominent harmonic stacks and minimal frequency modulation for FF and amplitude quantification. FF was calculated as the peak in the band-limited power spectrum of a 2–5 ms window within each syllable during which FF was stable. We measured syllable amplitude by detecting the peak of the smoothed (2 ms moving window) rectified audio waveform.

**Song similarity analysis**. To quantify the similarity between tutored or untutored songs and the tutor song (Supplementary Fig. 3), we applied a method for automatically classifying vocalizations in an unbiased and unsupervised fashion[40] based on their acoustic content. Briefly, this method extracts syllables from a test song (e.g., from a tutored or untutored bird) and assembles a statistical model based on the syllables' acoustic content. Through comparison to similarly constructed statistical models of the reference song (e.g., tutor song), these models are then used to estimate both the amount of information present in the reference that is absent from the test song (unlearned content) and the amount of information present in the test song that is absent from the reference song (improvised content).

**Data availability**. Data sets generated and analyzed in this study are available from the corresponding author upon request. Code used for analysis is also available upon request.

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

## Acknowledgements

We thank Andrea Hasenstaub and Michael Stryker for commenting on drafts of the manuscript and members of the Brainard lab for providing input at every stage of the project. This work was supported by the Howard Hughes Medical Institute and NIH grants R01MH055987 and R01DC006636 (msb), by an NIH F32 NRSA award (mnm), and by an NSF predoctoral award (cjc).

## Author contributions

M.N.M. and M.S.B. conceived the project, M.N.M. designed and performed the experiments, C.Y.J.C. contributed to in vivo pharmacology experiments, M.N.M. analyzed the data, M.N.M. wrote the manuscript, and M.N.M. and M.S.B. edited the manuscript.

## Additional information

**Competing interests:** The authors declare no competing financial interests.

