## [Peer Review File · Nature Communications]

Editorial Note: This manuscript has been previously reviewed at another journal that is not operating a transparent peer review scheme. This document only contains reviewer comments and rebuttal letters for versions considered at Nature Communications. Parts of this peer review file have been redacted as indicated

REVIEWERS' COMMENTS:

Reviewer #1 (Remarks to the Author):

In their study, Miller and colleagues observe specific learning-induced changes in the connectivity of a specific song production nucleus in the Bengalese finch. Compared with their previous submission, this version of the manuscript is greatly improved on two fronts. First, the authors address the potential function of inhibitory changes within RA by pharmacologically manipulating inhibitory transmission both in vivo and in vitro. They find that these manipulations can result in gross changes in specific features of song production, which indicate that inhibitory connections within RA are useful for maintaining precise articulation during singing. Second, the authors include considerably more quantification, and the paper now presents a more complete account of their experimental results. As a result, I think that this work should be accepted at Nature Communications in its present form.

Reviewer #2 (Remarks to the Author):

[Redacted]

I believe that the paper is interesting and absolutely belong to Nature Communications, provided that the following point is adequately addressed.

Figure 3b, it is proposed that tutoring changes the relative frequency of different connectivity patterns. It is shown, for example, that reciprocal connections are not statistically distinguishable from chance in untutored birds but it is significantly enriched in tutored birds. Based on this observation, the authors claim that tutoring enriches reciprocal connections. However, I believe that the current norm of the field for this type of claim requires that the statistical significance shown for the direct comparison between untutored vs. tutored. The argument that A is significant and B is not, so A and B are different, is not a strong argument. The same applies to the PN to FS connections.

Dear Dr. Ranade and Reviewers,

We thank the reviewers for again considering our manuscript and providing insightful comments and feedback. We are pleased that both reviewers found significant merit and interest in our revised submission and felt that our revisions addressed their concerns. Reviewer 2 requested that we perform additional statistical tests, which we agree are sensible and which further improve our manuscript. The reviewers' comments and our response to them are reproduced in full below.

Thank you again for the time and consideration you gave our manuscript, and for suggesting revisions that significantly expanded and improved our study from the initial submission.

Sincerely,

Michael and Mark

Reviewer #1:

In their study, Miller and colleagues observe specific learning-induced changes in the connectivity of a specific song production nucleus in the Bengalese finch. Compared with their previous submission, this version of the manuscript is greatly improved on two fronts. First, the authors address the potential function of inhibitory changes within RA by pharmacologically manipulating inhibitory transmission both in vivo and in vitro. They find that these manipulations can result in gross changes in specific features of song production, which indicate that inhibitory connections within RA are useful for maintaining precise articulation during singing. Second, the authors include considerably more quantification, and the paper now presents a more complete account of their experimental results. As a result, I think that this work should be accepted at Nature Communications in its present form.

We are pleased that our revisions thoroughly addressed the Reviewer's concerns about our original submission, and we appreciate the Reviewer's suggestions that prompted us to

expand our study to address FSI function in singing birds.

Reviewer #2:

[Redacted]

I believe that the paper is interesting and absolutely belong to Nature Communications, provided that the following point is adequately addressed.

Figure 3b, it is proposed that tutoring changes the relative frequency of different connectivity patterns. It is shown, for example, that reciprocal connections are not statistically distinguishable from chance in untutored birds but it is significantly enriched in tutored birds. Based on this observation, the authors claim that tutoring enriches reciprocal connections. However, I believe that the current norm of the field for this type of claim requires that the statistical significance shown for the direct comparison between untutored vs. tutored. The argument that A is significant and B is not, so A and B are different, is not a strong argument. The same applies to the PN to FS connections.

The Reviewer is correct that the statistical tests in Figure 3b compare the frequency of connection motifs found in each tutoring condition to motif frequencies in theoretical null distributions, rather than to frequencies in the other tutoring condition. We feel this comparison appropriately tests whether connections within RA are randomly organized, or whether there is instead some structure. Our finding that FSI connections in tutored but not untutored birds preferentially form reciprocal motifs supports our conclusion that these motifs are enriched, relative to randomly organized connections, after tutoring. However, we agree that it is useful to additionally test whether there are proportionally more reciprocal motifs after tutoring, as the Reviewer suggests, and as is indicated by Figure 3a. We tested this by comparing the observed frequency of each motif in our tutored dataset with multinomial

models built with each motif's observed likelihood in our untutored data. In this analysis, deviations from the model's predicted frequencies indicate enrichment or reduction of motifs after tutoring. We found that the frequency of all three connection motifs in tutored birds differed from the model's predictions: both FSI ↔ PN and PN → FSI connections were significantly enriched ($p < 0.005$) and FSI → PN connections were significantly depleted ($p < 0.00001$). These results are added to both the main text and Figure 3's legend, and the methods are revised accordingly:

Bold-Italicized text added to Results line 136-137:

“However, in tutored birds, reciprocal FSI ↔ PN patterns were present at more than double the probability expected by chance (multinomial test $p < 0.01$, Fig. 3B, right), ***and the proportion of FSI ↔ PN connections among all connections in tutored birds was significantly greater than in untutored birds (multinomial test $p < 0.005$).***”

Bold-Italicized text added to Figure 3 legend:

“Frequency of 1st-order connection patterns that we detected in untutored (left, red) and tutored (right, blue) birds relative to each connection pattern's expected frequency given unidirectional FSI → PN and PN → FSI connection probabilities in untutored and tutored birds. Each connection pattern was present at chance levels before tutoring, whereas reciprocal FSI ↔ PN connections were 2.39 times more frequent than predicted by random connectivity after song learning (multinomial test $p < 0.01$, 95% confidence interval = 1.68-3.11 fold, see Methods). Correspondingly, one-way PN → FSI connections were 0.41-fold less frequent than predicted after song learning (multinomial test $p < 0.05$). ***Relative to untutored birds, tutoring significantly increased the prevalence of FSI ↔ PN and PN → FSI connections***

(multinomial test $p < 0.005$) and reduced the prevalence of FSI→PN connections

(multinomial test $p < 0.00001$). Error bars indicate 95% confidence intervals. Connection pattern sample sizes are parenthetical.”

Bold-Italicized text added to Methods lines 357-360:

“To calculate the likelihood of observing reciprocal FSI-PN connections at the rate present in our datasets, we simulated tutored and untutored networks constructed with each dataset's unidirectional connection probabilities and sample size 100,000 times. Conceptually, this approach extends a binomial test to an arbitrary number of potential outcomes (in our case, the 4 possible connection motifs). ***To test whether motifs in tutored birds were significantly more or less common than in untutored birds, we created models with the frequencies of each connection pattern (FSI→PN, PN→FSI, FSI↔PN) present in untutored birds, and calculated the likelihood of observing the frequencies present in tutored birds.*** We validated our models with Matlab's mnpdf() and mnrnd() functions.”